# State-of-the-Art Evidence for Clinical Outcomes and Therapeutic Implications of Janus Kinase Inhibitors in Moderate-to-Severe Ulcerative Colitis: A Narrative Review

**DOI:** 10.3390/ph18050740

**Published:** 2025-05-17

**Authors:** Yunseok Choi, Suhyun Lee, Hyeon Ji Kim, Taemin Park, Won Gun Kwack, Seungwon Yang, Eun Kyoung Chung

**Affiliations:** 1Department of Pharmacy, College of Pharmacy, Kyung Hee University, Seoul 02447, Republic of Korea; cys2013@khu.ac.kr (Y.C.); sh198410@khu.ac.kr (S.L.); u00u96@khu.ac.kr (H.J.K.); ptyoon1@gmail.com (T.P.); 2Department of Pharmacy, College of Pharmacy, Woosuk University, Wanju 55338, Republic of Korea; 3Department of Regulatory Science, Graduate School, Kyung Hee University, Seoul 02447, Republic of Korea; 4Institute of Regulatory Innovation through Science, Kyung Hee University, Seoul 02447, Republic of Korea; 5Division of Pulmonary, Allergy and Critical Care Medicine, Kyung Hee University Hospital, Seoul 02447, Republic of Korea; wongunnim@naver.com; 6Department of Pharmacy, Kyung Hee University Hospital at Gangdong, Seoul 05278, Republic of Korea; 7Kyung Hee East-West Pharmaceutical Research Institute, Kyung Hee University, Seoul 02447, Republic of Korea

**Keywords:** ulcerative colitis, Janus kinase inhibitors, upadacitinib, tofacitinib, filgotinib, narrative review

## Abstract

**Background/Objectives**: Ulcerative colitis (UC) is a chronic inflammatory bowel disease characterized by relapsing inflammation and incomplete response to conventional therapies. Although biologics have advanced UC management, many patients with moderate-to-severe disease experience treatment failure, relapse, or adverse effects. This review evaluates the pharmacology, efficacy, and safety of oral Janus kinase (JAK) inhibitors—tofacitinib, upadacitinib, and filgotinib—to guide their clinical use in UC. **Methods**: A comprehensive literature review was conducted using the PubMed, Embase, Cochrane, and Web of Science databases to identify relevant studies on JAK inhibitors in UC. The review included Phase 3 randomized controlled trials (RCTs), real-world observational studies, and recent network meta-analyses. We assessed pharmacologic profiles, clinical efficacy, and safety data for tofacitinib, upadacitinib, and filgotinib. Additionally, we reviewed emerging pipeline agents and future directions in oral immunomodulatory therapy for UC. **Results**: All three agents demonstrated efficacy in the induction and maintenance of remission. Upadacitinib showed superior performance, including rapid symptom control, high clinical remission rates, and favorable long-term outcomes in both biologic-naïve and -experienced patients. Tofacitinib offered strong efficacy, particularly in early response, but was associated with higher risks of herpes zoster and thromboembolic events. Filgotinib provided moderate efficacy with a favorable safety profile, making it suitable for risk-averse populations. Meta-analyses consistently ranked upadacitinib highest in clinical efficacy and onset of action. **Conclusions**: JAK inhibitors offer effective and convenient oral treatment options for moderate-to-severe UC. Upadacitinib emerges as a high-efficacy agent; tofacitinib and filgotinib remain valuable based on patient-specific risk profiles. Future studies are needed to clarify optimal sequencing, long-term safety, and the role of emerging agents or combination therapies.

## 1. Introduction

UC is a subtype of inflammatory bowel disease (IBD). It is characterized by chronic inflammation of the colonic mucosa with recurrent episodes of relapse and remission, often requiring lifelong treatment to control disease activity [1,2]. Globally, several million patients were affected by UC, with colectomy required in approximately 20% of patients and hospitalization for one-third of patients with UC [3,4]. Although the pathogenesis of UC is not fully understood, it is believed to result from a confluence of genetic predisposition and environmental triggers, culminating in an abnormal immune response toward intestinal microbiota [5]. Historically, UC has been primarily managed with aminosalicylates, corticosteroids, and immunomodulators such as azathioprine, mercaptopurine, and methotrexate [6,7]. While conventional therapy can provide symptomatic relief, many patients fail to achieve long-term remission or adequate symptom control, potentially leading to the development of moderate-to-severe UC.

Over the past two decades, advanced therapeutic options such as biologic and targeted therapy for moderate-to-severe UC have been expanded by the introduction of tumor necrosis factor-alpha (TNF-α) inhibitors such as infliximab, which was approved as the first biologic therapy in 2005 [8,9,10,11,12,13]. These advanced therapies have been extensively evaluated in previous clinical studies, supporting the early use rather than gradual escalation of therapy after the failure of conventional treatment (i.e., 5-aminosalicylates) to improve treatment outcomes [4]. Despite this paradigm shift in the treatment of moderate-to-severe UC as recommended in the American Gastroenterological Association (AGA) Living Clinical Practice Guideline (2024) [4], a substantial proportion of patients showed no response at all or a gradual loss of response over time with reported rates of relapse ranging from 67% to 83% after 10 years of advanced therapy [14]. Indeed, a recent systematic review and meta-analysis reported the annual loss of treatment response to infliximab and adalimumab as 10% and 13%, respectively, with a higher risk typically observed within the first year of treatment [15]. Dose escalation might often improve treatment outcomes in patients with inadequate response [16]; however, a change in the selected medication might ultimately be warranted for the individual patient, further complicating the long-term management of UC [4]. Challenges remain for treating moderate-to-severe UC with advanced therapies to optimize treatment outcomes in real-world clinical practice, including high costs, extra monitoring for efficacy and safety, and less-preferred routes of parenteral administration [17].

Among advanced therapies, Janus kinase (JAK) inhibitors are considered a promising therapeutic alternative for moderate-to-severe UC as orally administered medications [18]. They target the Janus kinase-signal transducer and activator of transcription (JAK-STAT) pathway, which is critical for cytokine signaling to regulate multiple immune processes implicated in UC pathogenesis [19]. The JAK-STAT pathway is activated by various cytokines, growth factors, and other specific molecules, provoking numerous pro-inflammatory cytokines, including type I interferons and interleukins (IL) such as IL-2, IL-4, IL-6, IL-12, and IL-23 [20,21,22,23]. The human JAK family includes at least four intracellular tyrosine kinases, JAK1, JAK2, JAK3, and tyrosine kinase 2 (TYK2), interacting with STAT proteins to mediate downstream signaling, ultimately orchestrating diverse physiological events ranging from immune modulation and hematopoiesis to tissue repair [23,24,25,26,27,28] (Figure 1).

All currently approved JAK inhibitors predominantly target the JAK1-mediated pathway in UC pathogenesis with varying selectivity against specific JAK isoforms (i.e., JAK1, JAK2, JAK3, and TYK2) [29,30,31,32]. First-generation JAK inhibitors, such as tofacitinib, non-selectively inhibit multiple JAK isoforms (JAK1, JAK2, and JAK3), thereby increasing off-target effects [33]. Second-generation agents, including upadacitinib and filgotinib, exhibit greater selectivity against JAK1, potentially reducing safety concerns. The difference in isoform specificity among JAK inhibitors might account for the tolerability and efficacy of each agent with an increased risk for anemia and other hematologic adverse events resulting from substantial JAK2 inhibition of tofacitinib compared to newer-generation agents [34,35]. Considering the role of JAK1 in UC pathogenesis by regulating pro-inflammatory cytokines, notably IL-6, IL-12, and IL-23 [36,37], JAK inhibitors are an effective treatment option for moderate-to-severe UC. However, despite ongoing progress in medicinal chemistry, developing highly selective JAK inhibitors remains challenging due to the marked structural homology among JAK isoforms, particularly in the ATP-binding sites of their tyrosine kinase (JH1) and pseudokinase (JH2) domains [21,38,39,40]. Such similarity makes it difficult to target a single isoform without inducing off-target effects, underscoring the need to strike a careful balance between therapeutic efficacy and safety during JAK inhibitor development. Whereas first- and second-generation inhibitors predominantly act on the ATP-binding region of the JH1 domain, novel agents are emerging toward the JH2 domain to improve isoform specificity [21,38,39,40]. By selectively modulating JH2, these newer compounds aim to preserve efficacy while lessening the off-target risk associated with JH1-directed drugs. Such an approach underpins the development of agents like deucravacitinib (TYK2) and Z583 (JAK3) to optimize the balance between therapeutic benefit and safety [39,40]. Recent progress has led to alternative strategies for inhibiting JAK activity, including the use of allosteric compounds that target the JH2 domain, such as deucravacitinib (TYK2) [41], and irreversible covalent inhibitors like Z583, which selectively binds a distinct cysteine residue (Cys909) in JAK3 [42]. Although Z583 is still in preclinical development for rheumatoid arthritis (RA) rather than for IBD, its pharmacological mechanism underscores the ongoing efforts to improve isoform specificity and minimize systemic immunosuppression. A number of next-generation JAK inhibitors are also being explored in autoimmune and inflammatory disorders, including UC. Currently, all of the JAK inhibitors approved by global regulatory agencies are indicated for managing moderate-to-severe active UC; however, the specific treatment indications differ between the FDA and EMA. The U.S. Food and Drug Administration (FDA) restricts JAK inhibitor use to adult patients with an inadequate response, loss of response, or intolerance to at least one TNF-α inhibitor, thereby supporting their positioning as post–TNF antagonist therapeutic options [38,43]. In contrast, the European Medicines Agency (EMA) allows broader use in patients who have failed or are intolerant to conventional therapies, including corticosteroids, immunomodulators, or biologics, without requiring prior TNF-α inhibitor exposure [44,45,46,47]. This distinction reflects more restrictive labeling under the FDA than the more flexible and inclusive indication permitted by the EMA in patients with an inadequate response or intolerance to at least one TNF-α inhibitor. However, according to recently published clinical studies [48,49,50], the potential use of JAK inhibitors as a first-line agent for biologic-naïve patients with moderate-to-severe UC has been emerging. The AGA guideline for moderate-to-severe UC (2024) suggested JAK inhibitors as high- or intermediate-efficacy medications with a note for the FDA labels recommending their use in patients previously treated with at least one TNF antagonist [4]. Therefore, as suggested in the current clinical practice guideline as a knowledge gap, the therapeutic place of JAK inhibitors for moderate-to-severe UC might need to be further elaborated in various patient populations, including biologic-naïve and biologic-experienced individuals. Our objectives were to review comprehensive clinical and scientific evidence, specifically including the state-of-the-art literature, for currently approved JAK inhibitors (i.e., tofacitinib, upadacitinib, filgotinib) regarding their clinical pharmacology, efficacy/effectiveness, and safety and to suggest comparative therapeutic implications of JAK inhibitors in moderate-to-severe UC. Ultimately, this review might assist clinicians in making evidence-based, informed treatment decisions.

## 2. Literature Search Strategy

A comprehensive narrative review was undertaken to compile current evidence on the clinical pharmacology, efficacy/effectiveness, and safety of the oral JAK inhibitors including upadacitinib, tofacitinib, and filgotinib for moderate-to-severe UC. Searches were conducted in four major databases—PubMed, Embase, the Cochrane Central Register of Controlled Trials (CENTRAL), and Web of Science—spanning their inception to 15 March 2024.

The search strategy combined the following Medical Subject Headings and free-text terms: “ulcerative colitis”, “JAK inhibitors”, “tofacitinib”, “upadacitinib”, “filgotinib”, “Janus kinase”, “JAK-STAT”, “clinical trial”, “real-world evidence”, and “meta-analysis”. Boolean operators (AND/OR) were applied to refine the scope. Studies were included if they were pivotal Phase 3 randomized controlled trials, observational real-world studies, or systematic reviews and network meta-analyses assessing the clinical or pharmacological impact of JAK inhibitors in moderate-to-severe UC. Articles published in languages other than English, as well as case reports, editorials, and non-peer-reviewed commentaries, were excluded. Additionally, studies were excluded if moderate-to-severe UC was not the primary indication for the clinical investigation, pediatric populations were enrolled, or JAK inhibitors were not assessed as an intervention. Proceedings from relevant conferences and reference lists in retrieved studies were also reviewed manually to ensure completeness.

In total, 465 records were initially retrieved. After removing duplicate records, 298 unique entries were screened based on their titles, abstracts, and full texts. Of these, 105 were excluded, mainly because they reported Phase 2 trials, assessed interventions other than JAK inhibitors, or enrolled patients with diseases other than UC, such as Crohn’s disease. Ultimately, 99 studies met the inclusion/exclusion criteria and are cited in this narrative review.

## 3. Clinical Pharmacology of JAK Inhibitors: Pharmacokinetics and Pharmacodynamics

### 3.1. Absorption and Bioavailability

All of the three approved JAK inhibitors (i.e., tofacitinib, upadacitinib, and filgotinib) are well absorbed from the gastrointestinal tract to systemic circulation after oral dosing. Tofacitinib generally reaches its peak plasma concentration (Cmax) in 9 to 14 h after oral administration with an oral bioavailability of approximately 74% [43,46]. It is a P-glycoprotein (P-gp) substrate with low cell membrane permeability; the co-administration of moderate-to-strong P-gp inhibitors such as cyclosporine substantially increased systemic exposure to tofacitinib, requiring dosage adjustments of tofacitinib in clinical practice [51]. For upadacitinib, the typical time to reach Cmax (Tmax) ranges from 2 to 4 h after oral administration with a higher reported bioavailability of approximately 79% [38,45,52,53]. It is also a substrate for P-gp and BCRP efflux transporters, but the inhibition of these transporters does not meaningfully alter upadacitinib exposure due to its high solubility and permeability. Therefore, clinically relevant drug–drug interactions via P-gp inhibition are not expected, unlike tofacitinib [54]. Filgotinib reaches its Cmax in approximately 2 to 3 h after administration, with the majority of the administered dose present as an active metabolite (GS-829845) systemically; GS-829845 primarily accounts for the circulating pharmacological activity of filgotinib [44,55]. No clinically significant food effects were observed for tofacitinib, upadacitinib, or filgotinib, allowing for flexible administration schedules regardless of meal timing [38,43,44,45,46].

### 3.2. Metabolism and Elimination

Tofacitinib is predominantly eliminated by hepatic metabolism (70%), with the remaining fraction (~30%) excreted unchanged in urine [43,46]. Upadacitinib is substantially metabolized in the liver (34%); it is also eliminated through feces (38%) and urine (24%) as an unchanged drug [38,45]. The hepatic metabolism of both tofacitinib and upadacitinib is primarily mediated by CYP3A4 and, to a much lesser extent, by other CYP isoforms (e.g., CYP2D6) [38,43,45,46]. Consequently, the concomitant use of moderate-to-strong CYP3A4 inhibitors including systemic antifungals (e.g., voriconazole), immunosuppressants (e.g., cyclosporine), and grapefruit juice significantly increases plasma concentrations of tofacitinib and upadacitinib, warranting dose adjustments (e.g., reduced dose to approximately 50%) to achieve a comparable area under the plasma concentration–time curve (AUC) [51]. No active metabolites are formed for tofacitinib and upadacitinib through hepatic metabolism. In contrast, both filgotinib and its active metabolite formed by carboxylesterases, GS-829845, are largely cleared renally (~87% and 54%, respectively) with a minor fraction excreted fecally [56,57].

### 3.3. Impact of Renal and Hepatic Impairment

According to previous studies in patients with mild-to-moderate kidney impairment, no clinically significant alteration was observed in systemic exposures of tofacitinib and upadacitinib [58]. For filgotinib (EMA-approved only), however, plasma concentrations may increase twofold in patients with moderate renal impairment (CrCl 15 to <60 mL/min), requiring a dose reduction to 100 mg once daily (QD) [44]. In patients with severe renal impairment (CrCl 15 to <30 mL/min), dose reductions are recommended: 30 mg to 15 mg QD for upadacitinib, 10 mg twice daily (BID) to 5 mg BID (or 5 mg QD) for tofacitinib, and a reduced 100 mg QD for filgotinib [38,43,44,45,46,47]. In patients with end-stage renal disease, the use of upadacitinib or filgotinib is not generally recommended due to insufficient data; in contrast, tofacitinib might be used with caution at the lowest approved dose (i.e., 5 mg QD) if benefits outweigh risks [38,43,44,45,46,47]. For patients with impaired liver function, tofacitinib, and filgotinib can be used without dose adjustment in mild hepatic impairment with Child–Pugh class A [38,43,44,45,46]. According to the FDA label, upadacitinib should be used at a dose reduced by approximately 33% during induction therapy (45 mg to 30 mg) in patients with mild-to-moderate hepatic impairment (i.e., Child–Pugh class A or B), with a 15 mg dose recommended for maintenance in moderate-to-severe UC.

In contrast, the EMA label of upadacitinib does not recommend dose adjustment for patients with mild or moderate hepatic impairment [38,45]. In moderate hepatic impairment, dose reduction by 50% is recommended for tofacitinib; however, filgotinib may be used without dose adjustment at conventional doses [43,44,46]. Due to insufficient data, none of the three JAK inhibitors are recommended in patients with severely impaired liver function (i.e., Child–Pugh class C) or end-stage liver disease [38,43,44,45,46].

### 3.4. Selectivity and Mechanism of Action

Tofacitinib is considered a first-generation JAK inhibitor that targets JAK1, JAK2, and JAK3, thereby influencing a broad array of cytokine signaling pathways [32,59,60]. Due to this broader scope, the drug is effective across multiple immune-mediated conditions but may also lead to off-target effects, such as hematologic abnormalities at higher doses [59,61]. Alternatively, upadacitinib is classified as a second-generation JAK inhibitor with strong selectivity for JAK1 over JAK2, JAK3, and TYK2. This selectivity is intended to reduce off-target activity while maintaining potent inhibition of pro-inflammatory cytokines implicated in UC pathogenesis, including IL-6 and IL-23 [30,54,55]. Filgotinib, another second-generation agent, predominantly inhibits JAK1 while exhibiting moderate-to-minimal activity on JAK2, JAK3, and TYK2 [31,57,62,63]. In human cellular assays, filgotinib has shown the preferential inhibition of cytokine signaling through JAK1/3, JAK1/2, and JAK1/TYK2 complexes, with functional selectivity over JAK2/JAK2 or JAK2/TYK2 pathways. Notably, filgotinib’s primary active metabolite, GS-829845, is approximately tenfold less potent than the parent drug in vitro but retains a similar JAK1-selective profile [64]. Despite its lower potency, pharmacokinetic/pharmacodynamic (PK/PD) modeling and simulations from studies in healthy volunteers and RA patients suggest that systemic exposure to GS-829845 is sufficiently high to meaningfully contribute to the overall pharmacodynamic effects of filgotinib [65,66,67]. This difference in selectivity and active metabolite contribution may result from distinct binding kinetics and conformational affinity for the JAK1 catalytic domain across agents. Notably, concentration-dependent off-target effects have been more frequently reported with tofacitinib, where significant inhibition of JAK2 and JAK3 is observed at plasma concentrations exceeding 100 nM [68], correlating with hematologic toxicities in clinical settings. In contrast, upadacitinib and filgotinib demonstrate improved selectivity profiles, maintaining predominant JAK1 inhibition even at higher concentrations, thereby minimizing the risk of unintended immunosuppressive effects such as natural killer (NK) cell depletion or anemia. Such pharmacodynamic distinctions are expected to have downstream implications for safety monitoring and long-term management strategies in clinical practice [42,69]. To illustrate the isoform specificity of each agent, Table 1 presents in vitro IC₅₀ estimates for tofacitinib, upadacitinib, and filgotinib against JAK1, JAK2, JAK3, and TYK2. The in vitro specificity data highlight the broader inhibition of tofacitinib across multiple JAK isoforms with more pronounced selectivity against JAK1 for upadacitinib and filgotinib [42]. This difference can lead to important clinical implications for both efficacy and safety, considering higher JAK2 or JAK3 activity is frequently associated with an increased risk of hematologic abnormalities and other off-target effects [42]. All three drugs are administered orally, which may improve patient adherence and reduce the logistic challenges seen with intravenous or subcutaneous biologics [38,43,63]. Table 1 summarizes their key PK and PD properties.

### 3.5. Clinical Considerations

Overall, population PK studies indicate that body weight, gender, race, or age do not significantly alter exposure to upadacitinib, tofacitinib, and filgotinib [38,43,44,45,46]. Still, individual dose adjustments may be needed based on renal or hepatic function, possible drug–drug interactions, and other clinical factors such as concurrent immunosuppressive therapies or dietary factors. While PK and PD profiling offers foundational guidance for dose optimization, it does not fully reflect interindividual variability or capture long-term treatment durability in real-world clinical settings [19,48,70]. Patient-centered outcomes—including quality of life, fatigue, and symptom burden—may vary depending on comorbidities, prior biologic exposure, and disease severity [48,71,72,73]. Notably, differential PK/PD characteristics and JAK isoform selectivity among these agents highlight the need for stratified therapy in moderate-to-severe UC [19,30,33]. For example, tofacitinib requires cautious dosing in renal or hepatic impairment due to greater systemic metabolism and JAK2/3 activity [43,45,59,61], whereas filgotinib and upadacitinib demonstrate more favorable selectivity and renal clearance profiles [38,44,45,57]. These clinical differences underscore the need to contextualize pharmacologic data with findings from large-scale Phase 3 RCTs, real-world observational studies, and network meta-analyses. Such evidence can help refine patient-specific treatment decisions by integrating drug efficacy, safety, durability, and the onset of action. Therefore, the following sections will synthesize the key results from pivotal Phase 3 programs, real-world data, and meta-analyses to guide the therapeutic positioning of JAK inhibitors in UC management.

## 4. Phase 3 Efficacy Data for JAK Inhibitors in UC

### 4.1. Upadacitinib

Upadacitinib has been studied extensively in the U-ACHIEVE and U-ACCOMPLISH Phase 3 programs, comprising two induction studies (U-ACHIEVE Induction 1 [UC1], U-ACCOMPLISH Induction 2 [UC2]) and a maintenance study (U-ACHIEVE Maintenance 3 [UC3]) [74,75]. In the induction trials, patients with moderate-to-severe UC received upadacitinib 45 mg QD or placebo for 8 weeks [74,75]. Across both UC1 and UC2, upadacitinib yielded significantly higher clinical remission rates at week 8 compared to placebo (26.1% vs. 4.8% and 33.5% vs. 4.1%, respectively; both *p* < 0.001) (Table 2) [74,75,76]. Upadacitinib also demonstrated early efficacy, with a significantly higher proportion of patients achieving clinical response by week 2 (60% vs. 27% in UC1 and 63% vs. 26% in UC2, both *p* < 0.0001). Additional endpoints, such as histologic endoscopic mucosal improvement (HEMI) and mucosal healing, were significantly improved in the upadacitinib group compared with placebo [77]. In both induction studies, efficacy was evaluated across biologic-naïve and biologic-experienced populations. In biologic-naïve patients, clinical remission rates at week 8 were 35.2% vs. 9.2% (UC1) and 37.5% vs. 5.9% (UC2) for upadacitinib vs. placebo (*p* < 0.001). Among biologic-experienced patients, who had failed multiple biological therapies, remission rates were 17.9% vs. 0.4% (UC1) and 29.6% vs. 2.4% (UC2; all *p* < 0.001). Clinical response rates showed similar patterns of efficacy in both populations (biologic-naïve: 81.8% vs. 42.1% in UC1, 79.8% vs. 31.8% in UC2; biologic-experienced: 64.4% vs. 12.8% in UC1, 69.4% vs. 19.3% in UC2) [74,75,76]. In the maintenance study (UC3), responders from the induction phase were re-randomized to receive upadacitinib 15 mg or 30 mg QD or placebo [74,75]. At week 52, clinical remission rates were markedly higher in the upadacitinib groups (42% and 52%, respectively) compared with placebo (12%; *p* < 0.001) (Table 3.) [74,75]. Sustained mucosal healing and enhanced patient-reported outcomes (e.g., IBDQ, FACIT-F) were also observed [30]. In the maintenance study, clinical remission rates at week 52 were significantly higher in both biologic-naïve (43.9% for 15 mg, 54.0% for 30 mg vs. 17.6% placebo) and biologic-experienced populations (40.5% for 15 mg, 49.1% for 30 mg vs. 7.5% placebo; all *p* < 0.001). Endoscopic improvement showed similar trends, with higher rates in both upadacitinib doses compared to placebo across both patient populations [30].

### 4.2. Filgotinib

The efficacy of filgotinib was evaluated in the SELECTION trial, a Phase IIb/III study comprising biologic-naïve and biologic-experienced patients [31]. During a 10-week induction phase, patients were randomized to filgotinib 100 mg, 200 mg, or placebo QD [31]. Clinical remission rates at week 10 were significantly higher with filgotinib 200 mg than placebo in both biologic-naïve (26.1% vs. 15.3%, *p* = 0.01) and biologic-experienced patients (11.5% vs. 4.2%, *p* = 0.001) (Table 2) [31]. The 100 mg dose did not achieve statistical significance [31]. Endoscopic improvement, mucosal healing, and clinical response were also significantly higher in the 200 mg group compared to placebo [31]. Patients achieving response in the induction phase were re-randomized to filgotinib 100 mg, 200 mg, or placebo for 52 weeks [31]. At week 52, filgotinib 200 mg demonstrated higher clinical remission rates compared with placebo (23.8% vs. 13.6%, *p* = 0.042), with sustained improvements in mucosal healing and quality of life (Table 3) [31].

### 4.3. Tofacitinib

The OCTAVE clinical program evaluated tofacitinib in two induction trials (OCTAVE Induction 1 and 2) and a maintenance study (OCTAVE Sustain) [32,78]. In the induction studies, patients received tofacitinib 10 mg BID or placebo for 8 weeks. Clinical remission rates at week 8 were significantly higher in the tofacitinib group (18.5% vs. 8.2% in Induction 1 and 16.6% vs. 3.6% in Induction 2, both *p* < 0.001) (Table 2) [32,78]. Mucosal healing was observed in 31.3% and 28.4% of tofacitinib-treated patients, compared with 15.6% and 11.6% in placebo (both *p* < 0.001) [32,78]. In the OCTAVE Sustain trial, responders to induction therapy were re-randomized to receive tofacitinib 5 mg or 10 mg BID, or placebo, for 52 weeks. Clinical remission rates at week 52 were significantly higher in the 10 mg group than placebo (40.6% vs. 11.1%, *p* < 0.001), with the 5 mg dose also showing efficacy (34.3%, *p* < 0.001) (Table 3) [32,78]. Notably, the biologic-experienced population in these trials was limited to TNF-α inhibitor failures, unlike the more recent trials of upadacitinib and filgotinib, which included patients who had failed multiple biological mechanisms of action [32,78]. This difference in study populations should be considered when comparing efficacy rates across programs, as the OCTAVE trials potentially enrolled a less treatment-refractory population. In TNF-α inhibitor-naïve patients, clinical remission rates at week 8 were 26.2% vs.15.5% (Induction 1) and 21.7% vs. 7.7% (Induction 2) for tofacitinib vs. placebo. Among TNF-α inhibitor-experienced patients, remission rates were 11.1% vs. 1.6% and 11.7% vs. 0.0%, respectively [32,78]. In both trials, the treatment effect was similar between those who had received previous treatment with a TNF antagonist and those who had not [32]. Overall, tofacitinib demonstrated robust induction and maintenance efficacy in UC.

### 4.4. Phase 3 Safety Data for All JAK Inhibitors

In the induction phase, the overall incidence of adverse events was similar across the three JAK inhibitors: 53–56% with upadacitinib 45 mg QD, 53.6% with filgotinib 200 mg QD, and 54.1–56.5% with tofacitinib 10 mg BID. Serious adverse events were reported in 3–4% of patients across all three medications. The most common adverse events included nasopharyngitis (4–7%), headache (2–8%), and worsening UC (1–3%). (Table 4) During maintenance therapy, adverse event rates were higher but remained comparable among treatments: 78–79% with upadacitinib for both 15 mg and 30 mg; 66.8% with filgotinib 200 mg; and 72.2% and 79.6% with tofacitinib 5 mg and 10 mg, respectively. Discontinuation rates due to adverse events were notably higher with tofacitinib (9.1–9.7%) compared to upadacitinib (4–6%) and filgotinib (3.5%) (Table 4) [31,32,74,75,76,78]. Key safety concerns observed in these trials included infections, herpes zoster reactivation, thromboembolic events, and major adverse cardiovascular events (MACE). Any infection occurred in 35-40% of patients across all treatments, with serious infections more frequently reported with upadacitinib (3%) compared to filgotinib (1.0%) and tofacitinib (0.5–1.0%). Herpes zoster reactivation was most commonly observed with tofacitinib 10 mg (5.1%) and upadacitinib (4% in both doses), while filgotinib demonstrated the lowest rate (0.5%). Regarding thromboembolic events, venous thromboembolism was reported only with upadacitinib 30 mg (1%, two cases) during maintenance, while major adverse cardiovascular events were noted only with tofacitinib (one case each in 5 mg and 10 mg groups). Malignancy rates were generally low across treatments, with non-melanoma skin cancer reported only in upadacitinib 30 mg (1%) and tofacitinib 10 mg (three cases) during maintenance. These safety findings emphasize the importance of careful monitoring, particularly for infections and herpes zoster reactivation, and suggest that the choice of JAK inhibitor and dose should be individualized based on patient risk factors and comorbidities (Table 4 and Table 5) [31,32,74,75,76,78]. These findings highlight the need to carefully monitor infections, thromboembolic complications, and cardiovascular risks in patients receiving JAK inhibitors. Clinicians should individualize treatment and dosing according to each patient’s comorbidities and prior treatment history, ensuring optimal efficacy while minimizing adverse outcomes.

## 5. Real-World Effectiveness and Safety of JAK Inhibitors in UC

While Phase 3 trials provide high-quality evidence of efficacy and safety under controlled conditions, real-world data (RWD) complement these findings by reflecting outcomes in heterogeneous, treatment-refractory populations with varied clinical characteristics. Across the JAK inhibitor class—upadacitinib, tofacitinib, and filgotinib—accumulating RWD reinforces their therapeutic roles in UC while also highlighting important differences in early response, long-term durability, and safety profiles. Upadacitinib demonstrates rapid and robust real-world effectiveness in both biologic-naïve and biologic-experienced patients. In a U.S. cohort study, clinical remission rates exceeded 80% at week 8, with symptomatic improvement observed as early as week 2, even among patients previously exposed to tofacitinib [79]. In an international dataset of 357 patients, sustained clinical remission (59%) and symptomatic relief were maintained through 52 weeks, including among therapy-naïve individuals [80]. The rapid onset and remarkable treatment response of upadacitinib in treatment-refractory patients underscore its role as a front-line advanced therapy, though real-world safety signals of an increased risk of adverse events such as acne might warrant attention.

Tofacitinib has the longest real-world track record, with broad-range evidence across multiple countries and subtypes of UC. Meta-analysis data confirm short-term remission in over one-third of patients by week 8, with sustained response beyond 12 months in many cases [81]. Long-term follow-up (up to 3 years) demonstrated over 50% drug retention rates and successful recapture in relapsed patients through reinduction [82,83]. Tofacitinib is particularly valuable for patients with acute severe UC, those requiring flexible dosing, and those previously exposed to multiple advanced therapies [84,85]. However, herpes zoster and treatment discontinuations remain concerns, reinforcing the need for individualized safety monitoring.

Filgotinib, a novel JAK inhibitor, shows favorable effectiveness and an excellent safety profile in real-world cohorts. According to the UK and Japanese studies totaling over 600 patients, clinical remission rates ranged from 47% to 76% with persistence rates exceeding 60% at one year [86,87,88]. A recent multicenter head-to-head analysis demonstrated that upadacitinib achieved superior short-term effectiveness compared to filgotinib but with higher rates of adverse events (45.7% vs. 24.5%) [89]. These findings position filgotinib as a compelling option for biologic-naïve patients or individuals at high risk for adverse events, while upadacitinib might be suggested as a better alternative for rapid control in refractory cases.

Taken together, RWD supports the complementary strengths of each agent: upadacitinib for its early onset and effectiveness in both biologic-naïve and -experienced patients, tofacitinib for its long-term durability and reinduction potential, and filgotinib for its safety and therapeutic benefits in broad patient populations. However, large variability in study design, patient selection, and endpoints across real-world evidence limits direct comparisons. Thus, additional real-world studies are critically needed to (1) compare effectiveness and safety among various agents in diverse populations, (2) identify subgroup-specific treatment advantages, and (3) inform the optimal sequencing and positioning of JAK inhibitors in UC management.

## 6. Comparative Insights from Meta-Analyses for Advanced Therapies in UC

Previously published systematic reviews and network meta-analyses have compared the efficacy and safety of JAK inhibitors (upadacitinib, tofacitinib, filgotinib) with other advanced UC therapies such as TNF-α inhibitors, vedolizumab, and ustekinumab [48,49,50,70,71,90]. Instead of performing a new meta-analysis due to the substantial heterogeneity in recently published studies, we critically synthesized findings from high-quality existing meta-analyses—each exclusively incorporating primary RCT data—to provide comparative insights on clinical efficacy, the speed of onset, and safety. We have triangulated meta-analysis results with individual trial findings where possible for clarity and robustness. Common endpoints for UC were used such as clinical remission, endoscopic improvement, and speed of symptomatic relief (Table 6).

### 6.1. Induction and Maintenance Efficacy

Upadacitinib has consistently demonstrated superior efficacy in both the induction and maintenance phases of UC treatment across multiple high-quality network meta-analyses. These findings are consistent with results from pivotal Phase 3 trials (U-ACHIEVE Induction and U-ACCOMPLISH), in which upadacitinib achieved clinical remission rates of 26–29% at week 8 and sustained long-term response through week 52 [30]. In the meta-analysis by Lasa et al., upadacitinib 45 mg QD was suggested to be the most effective treatment for the induction of clinical remission with the highest surface under the cumulative ranking curve (SUCRA) value of 0.996. This performance significantly exceeded all other advanced therapies, including tofacitinib and filgotinib, with respective odds ratios of 2.84 and 4.49 for clinical remission [90]. The complementary analysis by Burr et al. reported similar results, showing upadacitinib as the highest-ranking agent for achieving clinical remission in both biologic-naïve and biologic-experienced patients, with a P-score of 0.99. Tofacitinib, as shown in the OCTAVE Inductions 1 and 2 trials, demonstrated clinical remission rates of 18.5% and 16.6% at week 8 as well as sustained benefit in the OCTAVE Sustain maintenance trial [32]. While suggested as a moderate efficacy tier in meta-analyses, tofacitinib remains an important option, particularly for patients who may benefit from early symptomatic control. Filgotinib, as assessed in the SELECTION trial, showed clinical remission rates of 26.1% in biologic-naïve and 11.5% in biologic-experienced patients, with favorable tolerability [31]. Although ranked lower for efficacy in network meta-analyses, its safety profile supports use in select populations [48].

Further supporting the consistent, superior efficacy of upadacitinib, Panaccione et al. applied an intent-to-treat (ITT) approach to maintenance phase outcomes, adjusting for the probability of induction response [49]. In this Bayesian network meta-analysis, upadacitinib ranked highest across most of the long-term endpoints, including clinical remission, clinical response, and endoscopic improvement, reinforcing its value as a durable therapeutic option [49]. In addition to RCT-based evidence, a 2024 systematic review and pooled meta-analysis by Niu et al. synthesized data from eight studies involving a total of 2818 patients with either UC or Crohn’s disease [70]. For patients with UC specifically, upadacitinib yielded a pooled clinical remission rate of 25.4% (95% CI: 17–36%) and a clinical response rate of 72.6% (95% CI: 69–76%). These findings further support the robust and consistent treatment response of upadacitinib, including in populations with prior treatment failures, thereby confirming its utility in both short- and long-term disease control strategies [70].

### 6.2. Onset of Action

The speed of therapeutic onset is a critical consideration in the management of moderate-to-severe UC, particularly for patients experiencing severe symptoms or corticosteroid dependence. In a Bayesian network meta-analysis by Attauabi et al., upadacitinib ranked first among all agents for both clinical response and clinical remission by week 2 of therapy [71]. Tofacitinib followed as the second-fastest-acting treatment, demonstrating its value in achieving early symptom relief [71]. These findings were corroborated by a frequentist network meta-analysis conducted by Ahuja et al., which evaluated symptomatic remission rates at weeks 2, 4, and 6 [50]. Upadacitinib was found to be significantly more effective than all other advanced therapies across all time points. The estimated proportion of patients achieving symptomatic remission at week 2 was 68% for upadacitinib, markedly higher than the rates observed with filgotinib (22%), vedolizumab and ustekinumab (approximately 16–18%), and ozanimod (10.9%) [50]. Taken together, these data strongly support the therapeutic roles of upadacitinib as the fastest-acting therapy currently available for moderate-to-severe UC. Its ability to induce rapid symptom relief offers a significant clinical advantage, especially in scenarios where timely disease control is paramount.

### 6.3. Comparative Safety Across JAK Inhibitors and Biologics

While differences in efficacy and onset of action are well established among JAK inhibitors, safety remains a critical dimension in therapeutic decision-making. Across six major network meta-analyses, no statistically significant differences were observed in the incidence of serious adverse events (SAEs) between JAK inhibitors—including upadacitinib—and biologic therapies. Although upadacitinib was associated with the highest overall adverse event rate among tested agents (SUCRA 0.843) in a meta-analysis by Lasa and colleagues, the incidence of SAEs remained comparable to those reported for other advanced therapies, indicating an acceptable safety profile [90]. Burr et al. reinforced these findings, showing that while tofacitinib was found to have a higher relative risk of infections (RR 1.41), upadacitinib was associated with a lower rate of treatment discontinuation due to adverse events when compared to placebo. This suggests that, despite its potency, upadacitinib maintains good tolerability in clinical practice [48]. The study by Panaccione et al. further highlighted that upadacitinib was associated with lower odds of treatment discontinuation due to adverse events during the induction phase while maintaining a safety profile similar to that of other agents across both induction and maintenance periods [49]. Consistent with these findings, the pooled meta-analysis by Niu et al. reported an SAE rate ranging from 6.0% to 7.8% for upadacitinib in patients with UC, further supporting its favorable benefit–risk balance [70].

### 6.4. Insights from Meta-Analyses and Clinical Implications

In summary, upadacitinib has emerged as the most effective option, particularly among patients who have failed prior biologic therapies, achieving high rates of clinical remission and endoscopic improvement [49,50,70]. Although tofacitinib demonstrates strong overall efficacy, its benefits appear most pronounced in biologic-naïve populations, where rapid symptom control is essential [48]. Filgotinib often ranks slightly lower in terms of pooled clinical remission but is distinguished by a favorable safety profile, making it a strategic option for patients who may be at greater risk for adverse events [48]. These meta-analyses also underscore notable differences in adverse-event profiles. Tofacitinib exhibits a higher incidence of infections (including herpes zoster) and thromboembolic events, especially at elevated doses [48,71]. Upadacitinib has been shown to have a generally manageable safety profile with a modestly elevated risk for herpes zoster, whereas filgotinib demonstrates the lowest rates of SAEs among the three agents [49,50]. Given these findings, tailoring therapy to individual patients is critical. Upadacitinib may be preferred in patients requiring robust disease control—especially those who have failed multiple biologics—owing to its superior efficacy and rapid onset of action [49,50]. Conversely, filgotinib could be advantageous in individuals who should avoid more aggressive immunosuppression due to comorbidities or advanced age, given its lower rates of serious infections [48]. Tofacitinib remains a valuable option, particularly in biologic-naïve patients who may benefit from its quick symptomatic relief [48]. In all cases, careful monitoring for infections, cardiovascular events, and other potential complications is warranted. Factors such as disease severity, prior treatment failures, and comorbid conditions (e.g., history of malignancy, thromboembolism) should guide drug selection and dose adjustments. As real-world data continue to expand, direct head-to-head comparisons and long-term follow-up data might further assist clinicians in refining the therapeutic place of JAK inhibitors in the treatment of moderate-to-severe UC, thereby optimizing clinical outcomes for a broad range of patient populations.

## 7. Limitations

Although this narrative review served to provide a comprehensive overview of the current evidence on JAK inhibitors in UC, including real-world experiences for the treatment of moderate-to-severe UC, several limitations should be noted. First, we did not conduct a quantitative meta-analysis; our conclusions instead rely on qualitative assessments drawn from available network meta-analyses, real-world evidence, and individual prospective trials. Second, the patient populations, study designs, and endpoints differed markedly across the included studies, especially regarding prior exposure to biologic agents and disease severity, making direct comparisons more difficult. Third, real-world data for filgotinib are still sparse, potentially resulting in underrepresentation of its broader clinical applications relative to upadacitinib and tofacitinib. Lastly, adverse event data included in this review were collected from prospective clinical trials and cohort data over a limited period of time; additional large-scale, long-term pharmacovigilance and real-world investigations may be warranted for definitive long-term safety outcomes including malignancy and cardiovascular risks.

## 8. Emerging JAK Inhibitors and Small Molecule Therapeutics in the IBD Pipeline

JAK1 is widely viewed as the key isoform in IBD due to its regulation of pro-inflammatory cytokines, notably IL-6, IL-12, and IL-23 [37,38]. Approved JAK inhibitors (upadacitinib, tofacitinib, and filgotinib) primarily exert their therapeutic effects through JAK1 inhibition [30,32,37]. Upadacitinib has demonstrated the highest degree of JAK1 specificity among approved agents, positioning it as a promising candidate for UC management [30]. The exact role of JAK2 remains more nuanced. While JAK2 inhibition can potentiate anti-inflammatory effects, potential off-target risks (e.g., anemia, thrombocytopenia) warrant caution [28,91]. The future development of more selective inhibitors may help optimize the balance between efficacy and safety, and combination therapies (e.g., JAK-STAT co-inhibition or JAK inhibitors in combination with biologics) may offer synergy in UC management.

Deucravacitinib, a highly selective TYK2 inhibitor, represents a promising class of oral agents that act upstream of the JAK-STAT pathway. Although currently approved for psoriasis, it is under investigation for IBD [41]. Its mechanism of binding to the TYK2 pseudokinase domain (as opposed to the ATP-binding site used by JAK1/2 inhibitors) offers a novel mechanism with potentially reduced off-target immunosuppression [92]. Povorcitinib (INCB054707), a selective JAK1 inhibitor, is also in early-phase clinical development for IBD, having shown immunomodulatory potential in other inflammatory diseases such as hidradenitis suppurativa [93]. Brepocitinib (PF-06700841) is an oral dual inhibitor of TYK2 and JAK1 currently under investigation for treating multiple immune-mediated diseases. Its dual-target mechanism enables the modulation of various cytokine signaling pathways, including IL-12, IL-23, and type I interferons [94], which are implicated in the pathogenesis of IBD. In early-phase clinical studies, induction therapy with brepocitinib in patients with moderate-to-severe active UC demonstrated superior efficacy compared to placebo, along with an acceptable short-term safety profile [95]. Furthermore, oral integrin inhibitors such as PN-943, a gut-restricted α4β7 integrin antagonist, have entered clinical trials and demonstrated early efficacy signals [92]. These agents represent an oral alternative to injectable biologics such as vedolizumab, with the additional convenience of oral administration and possibly fewer systemic effects [96,97].

Looking ahead, combination strategies involving JAK inhibitors with biologics, gut-selective S1P modulators, or even dual small-molecule regimens are being considered to maximize efficacy and durability while minimizing long-term safety concerns [98]. For example, co-targeting JAK1 and IL-23, or combining JAK inhibition with S1P modulation, may offer synergistic immunologic suppression while preserving mucosal integrity. As these novel agents advance through the pipeline, precision medicine approaches—such as biomarker-based patient selection using baseline CRP, genetic polymorphisms in JAK or TYK2, and molecular endotyping—are expected to enhance patient stratification and guide optimal therapeutic selection [99]. In conclusion, the therapeutic potential of JAK1 inhibition continues to expand with both approved and investigational agents. Simultaneously, a nuanced understanding of the role of JAK2 and the advent of alternative small molecules such as TYK2 inhibitors and oral integrin or S1P modulators open new frontiers in IBD treatment. The evolving pipeline underscores a shift toward convenient, effective, and safer oral therapies, with the potential for personalized and combinational treatment strategies to address the complex immunopathology of UC.

## 9. Conclusions

JAK inhibitors have broadened the treatment landscape for moderate-to-severe UC, offering oral and convenient alternatives to injectable biologics. Upadacitinib stands out for its potent and rapid efficacy in both induction and maintenance for various populations, including patients with and without prior biologic failure. Tofacitinib remains an efficacious option, particularly in biologic-naïve patients or those requiring flexible dosing strategies. Filgotinib, meanwhile, provides moderate efficacy with a favorable safety profile, making it a suitable choice for patients who need a conservative approach. While these agents address many unmet needs, questions remain regarding long-term treatment responses, head-to-head comparative efficacy, and patient-centered outcomes. Ongoing studies of novel JAK inhibitors, combination strategies, and biomarker-guided protocols will continue to refine personalized care in UC.

## Figures and Tables

**Figure 1 pharmaceuticals-18-00740-f001:**
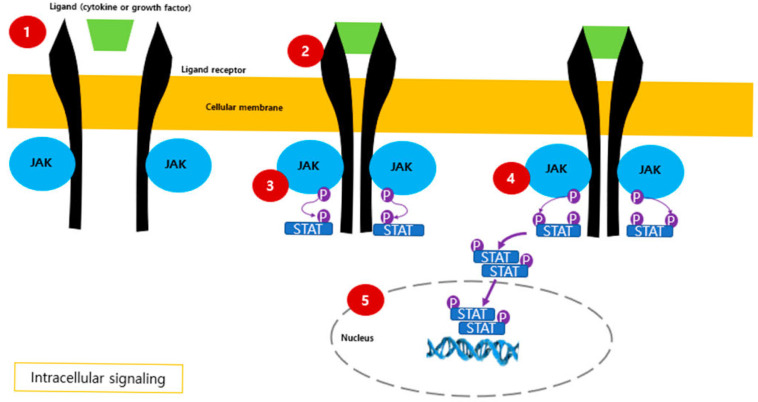
JAK-STAT pathway activation and intracellular signaling. Purple arrows represent the direction of intracellular signal transduction. Activation of the JAK-STAT signaling pathway: (1) A cytokine or growth factor binds to its specific cell surface receptor; (2) The receptor dimerizes and activates associated JAKs by phosphorylation; (3) JAKs phosphorylate tyrosine residues on the receptor, creating docking sites for STATs; (4) STATs are phosphorylated by JAKs, dissociate from the receptor, and form STAT dimers; (5) STAT dimers translocate to the nucleus, modulating gene transcription. Abbreviations: JAK, Janus kinase; STAT, signal transducer and activator of transcription; P, phosphorylation.

**Table 1 pharmaceuticals-18-00740-t001:** Pharmacokinetics and pharmacodynamics of upadacitinib, filgotinib, and tofacitinib.

PK/PD Characteristics		Upadacitinib		Filgotinib	Tofacitinib	
Administration Route		Oral		Oral	Oral	
IC_50_ (nmol/L) *		JAK1: 0.76 JAK2: 19 JAK3: 224 TYK2: 118		JAK1: 45 JAK2: 357 JAK3: 9097 TYK2: 397	JAK1: 15 JAK2: 71 JAK3: 45 TYK2: 472	
JAK Selectivity (Binding Domain)		JAK1 (JH1 domain)		JAK1 (JH1 domain)	JAK1, JAK2, JAK3 (JH1 domain)	
Time to Maximum Plasma Concentration (Tmax)		~2–4 h (median)		~2–3 h (median)	~9–14 h (median)	
Half-Life		~9–14 h		~7 h (parent) ~19 h (active metabolite)	~3 h	
Food Effects		Avoid grapefruit products (CYP3A4 inhibitors)		Not significantly affected by food	Avoid grapefruit products (CYP3A4 inhibitors)	
Bioavailability		~79%		~2.9% (parent) ~92% (active metabolite)	~74%	
Active Metabolite		Not known		GS-829845	Not known	
Urinary (Renal) Elimination		~24%		~87% (parent) ~54% (active metabolite)	~30%	
Fecal Elimination		~38%		~15% (parent) ~8.9% (active metabolite)	~0%	
Hepatic Metabolism		~34%		Not known	~70%	
Metabolic Enzymes		CYP3A4 (primary), CYP2D6 (minor)		CES2 (primary, intestinal), CES1 (minor, hepatic)	CYP3A4 (primary), CYP2C19 (minor)	
Prescribing Information Approved by		FDA	EMA	Only approved in EMA	FDA	EMA
Renal Impairment	Mild	No dose adjustment	No dose adjustment	No dose adjustment	No dose adjustment	No dose adjustment
	Moderate	No dose adjustment	No dose adjustment	100 mg QD (15 to <60 mL/min)	From 10 mg BID to 5 mg BID From 5 mg BID to 5 mg QD	No dose adjustment
	Severe	Induction: 30 mg QD Maintenance: 15 mg QD Not studied in ESRD (<15 mL/min)	Induction: 30 mg QD Maintenance: 15 mg QD Not studied in ESRD (<15 mL/min)	100 mg QD (15 to <60 mL/min) Not studied in ESRD (<15 mL/min)	From 10 mg BID to 5 mg BID From 5 mg BID to 5 mg QD Maintain reduced dose post-dialysis	From 10 mg BID to 5 mg BID From 5 mg BID to 5 mg QD Maintain reduced dose post-dialysis
Liver Disease	Mild	Induction: 30 mg QD Maintenance: 15 mg QD	No dose adjustment	No dose adjustment	No dose adjustment	No dose adjustment
	Moderate			No dose adjustment	From 10 mg BID to 5 mg BID	From 5 mg BID to 5 mg QD
	Severe	Not recommended	Not recommended	Not recommended	Not recommended	Not recommended

* IC₅₀ values represent the geometric mean of independent experiments carried out in the presence of 1 mmol/L ATP. Adapted from Dowty et al. [59]. Abbreviations: IC₅₀, half-maximum inhibitory concentration; JAK, janus kinase; TYK, tyrosine kinase; JH, tyrosine kinase domain; CYP, cytochrome P450; CES, carboxylesterase; ATP, adenosine triphosphate; FDA, food and drug administration; EMA, European Medicines Agency; QD, once daily; BID, twice daily; ESRD, end-stage renal disease.

**Table 2 pharmaceuticals-18-00740-t002:** Pivotal Phase 3 induction trials: clinical remission rates for JAK inhibitors (Week 8–10).

Drug	Trial (Year)	Population	Clinical Remission Rates at Week 8 to 10 ^2^		
			Placebo	Treatment (Dose)	Δ vs. Placebo
Upadacitinib	U-ACHIEVE (2022)	Bio-naïve 48% Bio-exp 52%	5%	26% (45 mg QD) *	21%
	U-ACCOMPLISH (2022)	Bio-naïve 51% Bio-exp 49%	4%	29% (45 mg QD) *	25%
Filgotinib ¹	SELECTION A (2021)	Bio-naïve 100%	15.3%	26.1% (200 mg QD) **	10.8%
	SELECTION B (2021)	Bio-exp 100%	4.2%	11.5% (200 mg QD) **	7.3%
Tofacitinib	OCTAVE 1 (2017)	TNFi-naïve: 47% TNFi- exp: 53%	8.2%	18.5% (10 mg BID) **	10.3%
	OCTAVE 2 (2017)	TNFi-naïve: 45% TNFi-exp: 55%	3.6%	16.6% (10 mg BID) *	13.0%

Statistical significance: * *p* < 0.001, ** *p* < 0.01. ¹ Filgotinib assessed at Week 10, others at Week 8. ^2^ Clinical remission defined as Mayo score ≤ 2, with stool frequency score ≤ 1 and not greater than baseline, rectal bleeding score = 0, and endoscopic subscore ≤ 1 without friability. Abbreviations: Bio-naïve, biologic-naïve; Bio-exp, biologic-experienced; TNFi-naïve, tumor necrosis factor inhibitor-naïve; TNFi-exp, tumor necrosis factor inhibitor-experienced; QD, once daily; BID, twice daily.

**Table 3 pharmaceuticals-18-00740-t003:** Pivotal Phase 3 maintenance trials: clinical remission rates for JAK inhibitors (Week 52).

Drug	Trial (Year)	Clinical Remission Rates at Week 52 ^1^		
		Placebo	Low Dose ^2^ (Δ vs. Placebo)	High Dose ^3^ (Δ vs. Placebo)
Upadacitinib	U-ACHIEVE (2022)	12%	42% (30%) *	52% (40%) *
Filgotinib	SELECTION (2021)	13.5%	23.8% (10.3%) **	37.2% (23.7%) *
Tofacitinib	OCTAVE Sustain (2017)	11.1%	34.3% (23.2%) *	40.6% (29.5%) *

Statistical significance: * *p* < 0.001, ** *p* < 0.05. ^1^ Clinical remission: Mayo score ≤ 2, with stool frequency score ≤ 1 and not greater than baseline, rectal bleeding score = 0, and endoscopic subscore ≤ 1 without friability. ^2^ Defined as 15 mg for upadacitinib, 100 mg for filgotinib, and 5 mg for tofacitinib. ^3^ Defined as 30 mg for upadacitinib, 200 mg for filgotinib, and 10 mg for tofacitinib. Δ: denotes differences between treatment groups.

**Table 4 pharmaceuticals-18-00740-t004:** Induction phase safety comparison.

Event Type	Upadacitinib 45 mg QD (% of Patients)	Filgotinib 200 mg QD ¹ (% of Patients)	Tofacitinib 10mg BID (% of Patients)
**Overall Safety**			
Any Adverse Events	U-ACHIEVE: 56% U-ACCOMPLISH: 53%	53.6%	OCTAVE 1: 56.5% OCTAVE 2: 54.1%
Serious Adverse Events	U-ACHIEVE: 3% U-ACCOMPLISH: 3%	4.3%	OCTAVE 1: 3.4% OCTAVE 2: 4.2%
D/C Due to AEs	U-ACHIEVE: 2% U-ACCOMPLISH: 2%	4.5%	OCTAVE 1: 3.8% OCTAVE 2: 4.0%
**Most Common AEs**			
Nasopharyngitis	4–5%		5–7%
Worsening UC	1–2%		2–3%
Headache	2–4%		7–8%
**Events of Interest**			
Any Infection		18.1%	OCTAVE 1: 23.3% OCTAVE 2: 18.2%
Serious Infection	1–2%	0.6%	OCTAVE 1: 1.3% OCTAVE 2: 0.2%
Herpes Zoster	<1%	0.2%	0.5–0.6%
VTE	0	0	0
MACE	0		0
Anemia	U-ACHIEVE: 1:3% U-ACCOMPLISH: 4%		

^1^ Combined data from induction studies. Abbreviations: QD, once daily; BID, twice daily; D/C, discontinuation; UC, ulcerative colitis; AEs, adverse events; VTE, venous thromboembolism; MACE, major adverse cardiovascular events.

**Table 5 pharmaceuticals-18-00740-t005:** Maintenance phase safety comparison.

Event Type	Upadacitinib QD (% of Patients)	Filgotinib 200 mg QD (% of Patients)	Tofacitinib BID (% of Patients)
**Overall Safety**			
Any Adverse Events	15 mg: 78% 30 mg: 79%	66.8%	5 mg: 72.2% 10 mg: 79.6%
Serious Adverse Events	15 mg: 7% 30 mg: 6%	4.5%	5 mg: 5.1% 10 mg: 5.6%
D/C Due to AEs	15 mg: 4% 30 mg: 6%	3.5%	5 mg: 9.1% 10 mg: 9.7%
**Most Common AEs**			
Nasopharyngitis	15 mg: 12% 30 mg: 14%		5 mg: 9.6% 10 mg: 13.8%
Worsening UC	15 mg: 13% 30 mg: 7%		5 mg: 18.2% 10 mg: 14.8%
**Events of Interest**			
Any Infection		35.1%	5 mg: 35.9% 10 mg: 39.8%
Serious Infection	15 mg: 3% 30 mg: 3%	1.0%	5 mg: 1.0% 10 mg: 0.5%
Herpes Zoster	15 mg: 4% 30 mg: 4%	0.5%	5 mg: 1.5% 10 mg: 5.1%
Malignancy ^1^	15 mg: <1% 30 mg: 1%	0.5%	5 mg: 0 ^2^ 10 mg: 0 ^2^
NMSC	15 mg: 0 30 mg: 1%	0	5 mg: 0 10 mg: 3 cases
VTE	15 mg: 0 30 mg: 2 cases (1%)	0	0
MACE	15 mg: 0 30 mg: 0		5 mg: 1 case 10 mg: 1 case
Anemia	15 mg: 5% 30 mg: 2%		

^1^ Excluding non-melanoma skin cancer (NMSC). ^2^ One case of breast cancer was reported in the maintenance placebo group. Abbreviations: QD, once daily; BID, twice daily; D/C, discontinuation; UC, ulcerative colitis; AEs, adverse events; NMSC, non-melanoma skin cancer; VTE, venous thromboembolism; MACE, major adverse cardiovascular events.

**Table 6 pharmaceuticals-18-00740-t006:** Systematic reviews and meta-analyses on biologics and small-molecule targeted therapies for UC.

Study	Sample Size/Objectives	Treatment Phase	Primary Outcomes	Subgorup Analysis (Bio-naïve/Bio-exp)	Investigated Agents	Highest-Rank Efficacy Agent	Highest-Rank Safety Agent	Key Implications
Lasa, J.S. et al. [90]	- 5904 patients from 29 trials - Evaluated efficacy and safety in moderate-to-severe UC	Induction	Clinical remission, endoscopic improvement	No	Upadacitinib Filgotinib Tofacitinib Infliximab Adalimumab Golimumab Etrolizumab Ustekinumab Vedolizumab Ozanimod	Upadacitinib	All comparable (N/S)	Upadacitinib with significant superiority to all other interventions for the induction of clinical remission
Burr, N.E. et al. [48]	- 12,504 patients from 28 trials - Evaluated efficacy and safety in moderate-to-severe UC	Induction	Failure to achieve clinical remission, failure to achieve endoscopic improvement	Yes	Upadacitinib Filgotinib Tofacitinib Infliximab Adalimumab Golimumab Etrolizumab Ustekinumab Vedolizumab Ozanimod	Upadacitinib for clinical remission, Infliximab for endoscopic improvement	All comparable (N/S)	Upadacitinib 45 mg QD ranked highest for clinical remission, infliximab 10 mg/kg ranked first for endoscopic improvement
Attauabi, M. et al. [71]	- 11,074 patients from 25 trials - Evaluated efficacy and safety in moderate-to-severe UC	Induction	Induction of clinical response and clinical remission in week 2	No	Upadacitinib Filgotinib Tofacitinib Infliximab Adalimumab Golimumab Ustekinumab Vedolizumab Ozanimod	Upadacitinib	Not available	Upadacitinib ranked highest for induction of clinical response and clinical remission in week 2: superior to all agents but tofacitinib
Panaccione, R. et al. [49]	- 23 trials - Evaluated efficacy and safety in moderate-to-severe UC	Induction and maintenance	Clinical response, clinical remission, endoscopic improvement	Yes	Upadacitinib Filgotinib Tofacitinib Infliximab Adalimumab Golimumab Ustekinumab Vedolizumab Ozanimod	Upadacitinib	All comparable (N/S)	Across all outcomes and regardless of prior biologic exposure, highest efficacy rates with upadacitinib
Ahuja, D. et al. [50]	- 14 RCTs - Comparatively evaluated symptomatic remission in week 2	Induction	Early symptomatic remission in week 2	Yes	Upadacitinib Filgotinib Infliximab Adalimumab Golimumab Ustekinumab Vedolizumab Ozanimod	Upadacitinib	Comparable safety profile (N/S)	Upadacitinib most effective in achieving symptomatic remission in week 2

Abbreviations: Bio-naïve, biologic-naïve; Bio-exp, biologic experienced; N/S, not significant; QD, once daily.

## Data Availability

All data generated or analyzed during this study are included in this published article. Additional datasets used during the narrative review are available from the corresponding authors upon reasonable request.

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
