# Peer review of "State-of-the-Art Evidence for Clinical Outcomes and Therapeutic Implications of Janus Kinase Inhibitors in Moderate-to-Severe Ulcerative Colitis: A Narrative Review"

_pharmaceuticals, 2025, doi:10.3390/ph18050740_

Round 1

Reviewer 1 Report

Comments and Suggestions for Authors

The narrative review by Choi et al. summarizes important data in the knowledge of pharmacology, efficacy, and safety of oral Janus kinase inhibitors—tofacitinib, upadacitinib, and filgotinib—to guide their clinical use in ulcerative colitis. I support its further processing after the clarification of the following concerns, specifically:

L55: “[1, 2]” – Please avoid the space delimitation when cite multiple references in brackets (in the line 62 is correct - [6,7]).

The introduction section is well written and effectively conveys the context and importance of the study.

The biggest problem of this manuscript is the fact that in its present form seems to be a large list of already well-known information, but without a discussion able to point up significant findings. The material and methods section is missing together with the description of the data mining process, the eligibility criteria and rejection criteria of the cited works, the number of chosen and rejected works, and keywords used in the data mining process and in the different databases. The total of the 99 articles mentioned in the reference list have been selected with a solid and clear methodology? If yes, what?

A paragraph with a complete and clear description of the bibliographic research is mandatory for a narrative review article in order to increase its value!

L249, 319, 355, 357, 496 – The design of Tables 1-6 is not in agreement with the journal's requirements. Furthermore, in Table 6 “Lasa JS, et al. (2022) [69]” must be “Lasa et al. [69]” – please carefully revise these issues.

L545: please provide a new version for the Conclusion section in a more condensed way. In its present form seems to be too long, without a concise summary of the key points addressed in the present review.

L600 and L612: Please be consistent in bolding or debolding the publication years of articles in the reference list.

Author Response

We have prepared a point-by-point response to the reviewers’ comments in the attached Word file. Kindly refer to the file for our detailed replies and revisions.

Point-by-point response to Comments and Suggestions for Authors

[Comment 1] L55: “[1, 2]” – Please avoid the space delimitation when cite multiple references in brackets (in the line 62 is correct - [6,7]).
[Response 1] We acknowledge the reviewer’s point. We have revised all citations throughout the manuscript to “[1,2]” format, eliminating the spaces between references (e.g., [1, 2]).

[Comment 2] The introduction section is well-written and effectively conveys the context and importance of the study.

The biggest problem of this manuscript is the fact that in its present form seems to be a large list of already well-known information, but without a discussion able to point up significant findings. The material and methods section is missing together with the description of the data mining process, the eligibility criteria and rejection criteria of the cited works, the number of chosen and rejected works, and keywords used in the data mining process and in the different databases. The total of the 99 articles mentioned in the reference list have been selected with a solid and clear methodology? If yes, what?

A paragraph with a complete and clear description of the bibliographic research is mandatory for a narrative review article in order to increase its value!

[Response 2] We appreciate the reviewer’s comment. As stated in the introduction section, this review was performed to further elaborate the therapeutic place of JAK inhibitors for moderate-to-severe UC in various patient populations including biologic-naïve and biologic experienced individuals. In doing so, we aimed to address the evolving regulatory landscape (e.g., EMA vs. FDA) and its implications for clinical practice, while also identifying current gaps in real-world application. In particular, we highlight emerging evidence supporting the potential first-line use of JAK inhibitors in biologic-naïve patients, an area that has not been fully explored in previous reviews. Our literature search strategy was established to meet this objective. To elaborate our literature search strategy in the manuscript according to the reviewer’s comment, we have added a new “Section 2. Literature Search Strategy” in the revised manuscript (Tracked version Page 04, Lines 181 – Page 05, Lines 209)

“2. Literature Search Strategy

A comprehensive narrative review was undertaken to compile current evidence on the clinical pharmacology, efficacy/effectiveness, and safety of the oral JAK inhibitors including upadacitinib, tofacitinib, and filgotinib for moderate-to-severe ulcerative colitis. Searches were conducted in four major databases—PubMed, Embase, Cochrane Central Register of Controlled Trials (CENTRAL), and Web of Science—spanning their inception to March 15, 2024.

The search strategy combined the following Medical Subject Headings and free-text terms: “ulcerative colitis,” “JAK inhibitors,” “tofacitinib,” “upadacitinib,” “filgotinib,” “Janus kinase,” “JAK-STAT,” “clinical trial,” “real-world evidence,” and “meta-analysis.” Boolean operators (AND/OR) were applied to refine the scope. Studies were included if they were pivotal Phase 3 randomized controlled trials, observational real-world studies, or systematic reviews and network meta-analyses assessing the clinical or pharmacological impact of JAK inhibitors in moderate-to-severe UC. Articles published in languages other than English, as well as case reports, editorials, and non-peer-reviewed commentaries, were excluded. Additionally, studies were excluded if moderate-to-severe UC was not the primary indication for the clinical investigation, pediatric populations were enrolled, or JAK inhibitors were not assessed as an intervention. Proceedings from relevant conferences and reference lists in retrieved studies were also reviewed manually to ensure completeness.

In total, 465 records were initially retrieved. After removing duplicate records, 298 unique entries were screened based on their titles, abstracts, and full texts. Of these, 105 were excluded, mainly because they reported Phase 2 trials, assessed interventions other than JAK inhibitors, or enrolled patients with diseases other than UC such as Crohn’s disease. Ultimately, 104 studies met the inclusion/exclusion criteria and are cited in this narrative review.”

[Comment 3] L249, 319, 355, 357, 496 – The design of Tables 1-6 is not in agreement with the journal's requirements. Furthermore, in Table 6 “Lasa JS, et al. (2022) [69]” must be “Lasa et al. [69]” – please carefully revise these issues.

[Response 3] We thank the reviewer’s comment. We reformatted all tables to comply with the journal’s style and format. In Table 6, references have been revised to the stated format, e.g., “Lasa JS et al. [69]”; first initials of first authors were maintained for efficient differentiation. Any similar issues have been addressed consistently across other tables.

[Comment 4] L545: please provide a new version for the Conclusion section in a more condensed way. In its present form seems to be too long, without a concise summary of the key points addressed in the present review.

[Response 4] We appreciate reviewer’s valuable suggestion. We have revised the Conclusion Section to be more concise, and essential clinical implications. (Tracked version Page 22, Lines 766 – 774)

“JAK inhibitors have broadened the treatment landscape for moderate-to-severe UC, offering oral and convenient alternatives to injectable biologics. Upadacitinib stands out for its potent and rapid efficacy in both induction and maintenance for various populations, including patients with and without prior biologic failure. Tofacitinib remains an effective option, particularly in biologic-naïve patients or those requiring flexible dosing strategies. Filgotinib, meanwhile, provides moderate efficacy with a favorable safety profile, making it a suitable choice for patients who need a conservative approach. While these agents address many unmet needs, questions remain regarding long-term positioning, head-to-head efficacy, and patient-centered comparative outcomes. Ongoing studies of novel JAK inhibitors, combination strategies, and biomarker-guided protocols will continue to refine personalized care in UC.”

[Comment 5] Please ensure consistency in bolding the publication years in the reference list.

[Response 5] We have revised all references for uniformity. The publication years are now in a consistent unbolded format.

Reviewer 2 Report

Comments and Suggestions for Authors

This work (Title: State-of-the-art Evidence for Clinical Outcomes and Therapeutic Implications of JAK Inhibitors in Moderate-to-Severe Ulcerative Colitis: A Narrative Review, Manuscript ID: pharmaceuticals-3598794) had established a literature review for the pharmacology, efficacy, and safety of oral Janus kinase (JAK) inhibitors (tofacitinib, upadacitinib, and filgotinib) to guide their clinical use in ulcerative colitis (UC), which conducted across PubMed, Embase, and Cochrane databases, including Phase 3 RCTs, real-world observational cohorts, and recent network meta-analyses. This reviewer is positive for publishing this manuscript if the authors appropriately revised the manuscript accordingly to the following comments.

Minor points:

  • “…in the current clinical practice guideline updated in 2024…”, this clinical practice guideline needs to see clear with “AGA Living Clinical Practice Guideline”.
  • Define abbreviation when they were first appeared in the title, abstract, early in manuscript, such as JAK and “ Introduction”.
  • The structure for JAK also need to discuss in “ Introduction”.
  • The JAK inhibitors which are investigated in preclinical and clinical phase need to show in “ Introduction”.
  • The JH1 or JH2 selective inhibitors targeting JAK need to be discuss.
  • There are not figures in this work. However, the structure of the JAK and JAK inhibitors can be shown with figure.
  • The activity for JAK inhibitors (tofacitinib, upadacitinib, and filgotinib) can be shown for JAK1, JAK2, JAK3, and TYK2 (maybe also for JH1 and JH2).
  • The meta-analyses were done with the original articles. However, the meta-analyses results have be compared in this work. This is not the best method for comparative insights for advanced therapies in US. The original articles have been analyzed and not the meta-analyses reulsts need to show in this work.
  • “A literature review was conducted across PubMed, Embase, and Cochrane databases, including Phase 3 RCTs, real-world observational cohorts, and recent network meta-analyses.” However, this method was not shown in this work. Meanwhile, the key database (Web of Science) is not considered in this work, which maybe missing important articles.
  • The limitation of this work needs to be considered.
  • It would be better if the manuscript is revised by professional English language editing services before publication.

Author Response

We have prepared a point-by-point response to the reviewers’ comments in the attached Word file. Kindly refer to the file for our detailed replies and revisions.

Round 2

Reviewer 1 Report

Comments and Suggestions for Authors

The authors correctly acknowledged all of my raised concerns.

Congratulation!

Reviewer 2 Report

Comments and Suggestions for Authors

The author has responsed all mine comments.